# The Janus Role of Adhesion in Chondrogenesis

**DOI:** 10.3390/ijms21155269

**Published:** 2020-07-24

**Authors:** Ignasi Casanellas, Anna Lagunas, Yolanda Vida, Ezequiel Pérez-Inestrosa, José A. Andrades, José Becerra, Josep Samitier

**Affiliations:** 1Institute for Bioengineering of Catalonia (IBEC), Barcelona Institute of Science and Technology (BIST), 08028 Barcelona, Spain; icasanellas@ibecbarcelona.eu (I.C.); jsamitier@ibecbarcelona.eu (J.S.); 2Department of Electronics and Biomedical Engineering, University of Barcelona (UB), 08028 Barcelona, Spain; 3Biomedical Research Networking Center in Bioengineering, Biomaterials, and Nanomedicine (CIBER-BBN), 28029 Madrid, Spain; andrades@uma.es (J.A.A.); becerra@uma.es (J.B.); 4Departamento de Química Orgánica, Universidad de Málaga-IBIMA, 29071 Málaga, Spain; yolvida@uma.es (Y.V.); inestrosa@uma.es (E.P.-I.); 5Centro Andaluz de Nanomedicina y Biotecnología-BIONAND, Campanillas, 29590 Málaga, Spain; 6Department of Cell Biology, Genetics and Physiology, Universidad de Málaga-IBIMA, 29071 Málaga, Spain

**Keywords:** dendrimer, nanopatterning, RGD, mesenchymal cell condensation, cell–cell interactions, YAP, chondrogenesis

## Abstract

Tackling the first stages of the chondrogenic commitment is essential to drive chondrogenic differentiation to healthy hyaline cartilage and minimize hypertrophy. During chondrogenesis, the extracellular matrix continuously evolves, adapting to the tissue adhesive requirements at each stage. Here, we take advantage of previously developed nanopatterns, in which local surface adhesiveness can be precisely tuned, to investigate its effects on prechondrogenic condensation. Fluorescence live cell imaging, immunostaining, confocal microscopy and PCR analysis are used to follow the condensation process on the nanopatterns. Cell tracking parameters, condensate morphology, cell–cell interactions, mechanotransduction and chondrogenic commitment are evaluated in response to local surface adhesiveness. Results show that only condensates on the nanopatterns of high local surface adhesiveness are stable in culture and able to enter the chondrogenic pathway, thus highlighting the importance of controlling cell–substrate adhesion in the tissue engineering strategies for cartilage repair.

## 1. Introduction

Articular cartilage is a connective tissue that covers the surface of bones at the joints, enabling them to slide over one another, reducing friction. The avascular and aneural nature of cartilage limits its ability of self-repair after injury, which frequently evolves to long-term lesions [1]. If the damage reaches the subchondral bone, the blood supply from the bone starts a healing process that results in the formation of a scar tissue made from fibrocartilage. Fibrocartilage is a more rigid structure than the original hyaline-like tissue and more prone to rupture under the tensions generated in the joint. It generally deteriorates over time with occasional progression to osteoarthritis.

Among the different techniques to treat severe cartilage lesions, autologous chondrocyte implantation (ACI) has gained a broad acceptance [2]. In ACI, chondrocytes are harvested and expanded from a small piece of cartilage from a biopsy of healthy tissue. After 1–3 months the expanded chondrocytes are implanted into the lesion and covered with a periosteal patch. Nevertheless, ACI is limited by the cell expansion step, which is required to generate constructs of a size large enough to fill the lesion. Chondrocytes dedifferentiate during monolayer culture expansion, with a progressive downregulation of chondrogenic genes and loss of chondrocyte morphology, adopting a fibroblast-like phenotype [2]. As an alternative to ACI, mesenchymal stem cells (MSCs) have been regarded as a promising cell source to tackle cartilage widespread lesions due to their vast proliferative capacity and differentiation potential. However, it seems that the natural lineage commitment of MSCs when stimulated for chondrogenesis leads to hypertrophic chondrocytes and endochondral bone formation [3]. Therefore, most current tissue engineering strategies address the early stages of MSC differentiation to direct chondrogenesis towards the formation of hyaline cartilage tissue and minimize hypertrophy [4].

Extracellular matrix adhesion plays a pivotal role in chondrogenesis, showing its influence from the initial cell aggregation into cell condensates [5]. Cell adhesion and differentiation are regulated at the nanoscale with features of similar size to the cell-adhesion receptors integrins interacting with them in the cells’ focal adhesions, thereby affecting downstream biochemical pathways and causing changes in cells morphology and fate [6]. Experiments using micellar lithography-based nanopatterning showed the relevance of the nanospacing between the cell adhesive motives and revealed a threshold value of around 70 nm on rigid substrates above which cell adhesion is significantly delayed [7,8]. Nanopatterned and nanostructured surfaces have long been applied as cell culture substrates to direct stem cell lineage commitment [9,10,11,12]. Although three-dimensional (3D) culturing such as alginate encapsulation, micromass and pellet cultures are the most widely applied culture systems for chondrogenesis in vitro [13], they show limitations in their ability to interrogate cells on cell-matrix interactions. Alternatively, two-dimensional (2D) culture is used. The effects of the nanospacing between the adhesive motifs in MSCs chondrogenic differentiation have been investigated on nanopatterns of the cell adhesive peptide arginine–glycine–aspartic acid (RGD). A large nanospacing (161 nm) between the RGD motifs led to a reduced spreading area and a higher chondrogenic induction [14]. In previous works, we used RGD dendrimer-based nanopatterns to evaluate the effects of local RGD surface density on cell adhesion and differentiation [15,16,17,18].

Here we applied RGD dendrimer-based nanopatterns to evaluate the effects of high and low local surface adhesiveness in directing MSCs chondrogenesis in vitro, focusing on the initial prechondrogenic condensation step. Nanopatterns of high local surface adhesiveness, with 90% of the surface area containing RGD nanospacings below 70 nm (S_90_), promoted cell adhesion and directional cell migration, thus favoring cell recruitment into the condensation sites. Adhesion also contributed to the mechanical stability of the condensates and compensated cell–cell interactions, allowing cells to enter the differentiation path. On the contrary, nanopatterns of low local surface adhesiveness, with only 18% of the surface area containing RGD nanospacings below 70 nm (S_18_), resulted in impaired condensation, which led to mechanically unstable condensates unable to proceed with the chondrogenic commitment. 

## 2. Results

### 2.1. Single-Cell Tracking: Cell Migration towards the Condensate Center

At the early stages of cartilage development, undifferentiated mesenchymal cells actively migrate to the condensation sites without cell proliferation [19]. The transforming growth factor (TGF) superfamily of proteins induces precartilage condensation by the regulation of the cell matrix glycoprotein fibronectin (FN), with the cells preferably accumulating in the regions of stronger cell-matrix adhesive interactions [5,20]. Here we used dendrimer-based nanopatterns of RGD to control the local surface adhesiveness of poly(L-lactic acid) (PLLA) substrates during chondrogenic differentiation [17]. Dendrimers adsorbed onto non-charged surfaces in a liquid-like order with a defined spacing [21]. The local surface adhesiveness of the nanopatterns was quantified by calculating the minimum interparticle distance (d_min_) and using a nanospacing of 70 nm as a threshold [7,8]. The percentage of surface area with d_min_ < 70 nm was measured and used to name the nanopatterns (Appendix A
Appendix A). Human mesenchymal stem cells (hMSCs) stained for the cell nuclei were seeded on nanopatterns of low (S_18_) and high (S_90_) local surface adhesiveness in chondrogenic medium and tracked by fluorescence live cell imaging during 40 h (Figure 1). Pristine non-patterned PLLA (S_0_) and FN-coated (S_FN_) substrates were the negative and positive controls, respectively. Selected time-lapse images, as seen in Figure 1a, showed that certain cells that adhered while still preserving their rounded morphology initiated mesenchymal cell condensation. Small groups of cells formed around these initial condensation centers in all the substrates within the first two hours after the cell seeding, except for the positive control. These groups of cells continuously explored the substrate by extending filopodia and coordinately moving to reach other groups of cells nearby. Nevertheless, only in S_90_ nanopatterns were outspread cell clusters clearly visible at the end of the experiment (t = 40 h). Cell cluster observation in the nanopatterns was determined by the initial number of cells adhered (Appendix A
Appendix A). In the positive control, cells adhered adopting a spindle-like morphology, consistent with a substrate-induced FN fibrillogenesis [22], and explored the substrate individually. The cell trajectories, depicted in Figure 1b, showed isotropic tracks in all the cases according to the uneven distribution of the adhesive motifs in the substrates. However, the ensemble of the tracks looked more compacted with the decreasing local surface adhesiveness, which indicated that cells in low adherent substrates barely moved away from the original location. This effect was highlighted in the quantification of the Euclidean distance, the shortest straight distance between start and end points in a cell’s trajectory, where the 99th percentile abruptly increased for the positive control (dashed red line in Figure 1c). The mean Euclidean distance was found not significantly different for S_0_ and S_18_ (16.8 ± 0.4 and 17.4 ± 0.3 μm, respectively), while it significantly increased for S_90_ and S_FN_ (21.0 ± 0.3 and 31.2 ± 0.4 μm, respectively). These results may lead to the assumption that cells moved less in low-adherence substrates. However, mean link displacement in the cell tracks, the distance covered by a cell between two consecutive frames, decreased with the increasing local surface adhesiveness (dashed blue line in Figure 1c). This means that cell movement was more directional with the increasing local surface adhesiveness.

### 2.2. Cell Condensates Growth and Compaction

As seen in the cell migration assays, prechondrogenic condensation of hMSCs resulted from extracellular matrix (ECM)-driven cell recruitment towards the center of condensations. The process of mesenchymal cell condensation encompasses precise cell shape deformations, which are primarily driven by changes in the organization of the actin cortex [23]. The actin cortex is a thin actomyosin network bound to the cell membrane that is found in most animal cells and that has a pivotal role in the mechanics of cell migration, cell division and morphogenesis [24]. A typical cortical actin disposition was observed in all the cell condensates (Figure 2a), while cells on S_FN_ substrates, arranged in a 2D configuration, showed a stress-fiber organization of the actin network [23]. Cell condensates require a critical size below which chondrogenesis will be anomalous or even discontinued, while excessively large condensates lead to aberrant tissue structures [19]. The cell viability assay indicated cells were alive in the condensates (Appendix A
Appendix A). Actin staining in the 3D confocal images was used to calculate the volume of the cell condensates after 6 h of chondrogenic induction (Figure 2b). We found that volume progressively increased with the increasing local surface adhesiveness, in agreement with previous results obtained from confocal area projections [17]. 

As the cell density increases in the condensation sites, intimate cell–cell contacts are established. During cell–cell contact formation, a local decrease in the interfacial cortex tension (cortical clearing) regulates the size of the adhesions between adjacent cells [25]. Areas of cortical clearing associated with cell–cell contacts in the condensates are magnified in Figure 2a. In mesenchymal cells, cell–cell contact is mediated by N-cadherins [26], and the surface tension of the forming tissue that originated from cortical contractions has been found to be proportional to the cadherins’ expression levels [27]. Therefore, to evaluate the condensates surface tension, we examined N-cadherin gene expression (*CDH2*). We found that the mRNA relative expression of *CDH2* decreased with the local surface adhesiveness of the substrate, with no significant differences between S_18_ and S_90_ condensates (Figure 2c). 

### 2.3. YAP-Mediated Mechanotransduction and Chondrogenic Commitment

MSCs fate is strongly influenced by the mechanochemical stimuli. Mechanical signals directing the cell fate involve molecular factors from the Hippo pathway [28]. The Hippo signaling pathway is a key regulator of organ size, controlling cell proliferation and apoptosis with an important role in stem cell and tissue specific progenitor cell self-renewal and differentiation. The core of this pathway is composed of a kinase cascade wherein the STE20 family kinases MST1 and MST2 in complex with the adaptor proteins SAV1, MOBKL1A and MOBKL1B phosphorylate and activate the NDR family kinases LATS1 and LATS2. Phosphorylated LATS1 and LATS2, in turn, phosphorylate and inactivate the transcriptional regulators TAZ and Yes-associated protein (YAP), sequestering them in the cytosol where they are degraded (inhibitory phosphorylation of YAP/TAZ) [29]. Therefore, when the Hippo pathway is inactive, YAP translocates into the cell nucleus promoting cell proliferation and stem cell self-renewal, and apoptosis is inhibited. Several upstream negative regulators of the Hippo pathway have been described. In particular, SRC kinase, which is activated by cell adhesion, facilitates YAP nuclear translocation by the SRC-mediated activating phosphorylation of YAP and the inhibitory phosphorylation of LATS1 [28,30]. Alterations in the YAP activity are the predominant mechanism by which mechanotransduction influences cell fate [31]. Therefore, to gain more insights on how local surface adhesiveness regulates chondrogenesis on the nanopatterns, we examined YAP localization (Figure 3).

We observed mostly cytoplasmic YAP in all the substrates (Figure 3a,b). Although mainly localized in the cytosol, the active YAP quantified in S_90_ condensates was significantly higher than in the rest of the substrates (Figure 3b), suggesting a certain mechanotransduction contribution in the YAP activity. Moreover, we found that the active YAP in the condensates increased after three days of chondrogenic induction, and that mRNA relative expression levels of the chondrogenic marker *SOX9* mirrored YAP in the different substrates (Figure 3c). *SOX9* was upregulated in S_90_ and downregulated in S_18_ and S_FN_, indicating that only cell condensates from S_90_ entered the chondrogenic path after 6 days of chondrogenic induction. During chondrogenesis, *SOX9* is co-expressed with *COL2A1*, the gene encoding type II collagen, the major cartilage ECM protein. Accordingly, type II collagen quantification showed significantly higher values for S_90_ (Appendix A
Appendix A).

## 3. Discussion

Here we studied the effects of RGD nanopatterns of low and high local surface adhesiveness in the prechondrogenic condensation of hMSCs. There is experimental evidence of the extracellular matrix organization at the nanoscale, such as the assembly of collagen fibers or the folding/unfolding of fibronectin [32,33,34], and of the subsequent organization of cell membrane receptors into nanoclusters [35]. The nanopatterning of ECM motifs for the study of cell-surface interactions at the nanoscale has highlighted the relevance of ECM ligand presentation to cells and demonstrated that cell adhesion is more affected by the local rather than the global ligand concentrations [36,37,38,39]. Experiments using micellar lithography-based nanopatterning showed the relevance of the nanospacing between the cell adhesive motives and revealed a threshold value of around 70 nm on rigid substrates above which cell adhesion is significantly delayed [7]. Since these first works in the field, others have emerged showing that not only cell adhesion, but also many other cell responses are affected by the nanoscale ligand presentation, including cell differentiation [8,9,10,11,12,40]. 

Previous works demonstrated the suitability of RGD dendrimer-based nanopatterns to sustain hMSCs adhesion and differentiation [16,17,18]. We found that chondrogenic differentiation is favored on nanopatterns of intermediate surface adhesiveness [17]. Results pointed out the relevance of the condensation phase in the subsequent differentiation events. Therefore, in this work we applied RGD dendrimer-based nanopatterns to the study of the effects of local surface adhesiveness in the initial prechondrogenic condensation step. Mesenchymal cell condensation is a prevalent morphogenetic transition regulated by cell adhesion. Therefore, and using the concept of developmental engineering, we aimed at exerting control on the early differentiation events to promote hMSCs commitment toward the formation of hyaline cartilage tissue and to minimize hypertrophy.

Chondrogenesis begins with cell recruitment to the condensation sites. Therefore, we first tracked individual cell trajectories on the nanopatterns and control substrates from the initial cell seeding. We observed that the formation of cell condensates on the nanopatterns is dependent on the availability of adhesion sites. A high local surface adhesiveness leads to a higher percentage of cells initially adhered, which when migrating shows a more directional movement.

Since the maintenance of front–rear asymmetry is essential for the directional persistence, we conclude that a fibrillar FN in the positive control could favor the spontaneous cell symmetry breaking and cytoskeletal polarization, thus initiating the persistent migration in the polarity direction [41]. In the absence of any preferential direction, a randomly migrating cell moves alternating fast translocation with slow rotation [42,43], which causes changes in orientation, as seen for the nanopatterns. The sustained periods of straight paths in the nanopatterns seem to be conditioned by the number of available adhesion sites. Therefore, in the low adhesive surfaces S_0_ and S_18_, the many changes in the migratory direction resulted in a lower total displacement (Euclidean distance), despite cells moving at a higher linear velocity (as indicated by larger mean link displacements in the same period of time).

A more directional movement favors cell–cell interactions, which are essential for cell condensations to evolve into more mature and stable prechondrogenic condensates. Cell condensation entails precise cell deformations, which are driven by the contractile tension gradients generated by the actomyosin cortex. Cortical contraction is a key determinant of surface tension, which is proportional to the expression of cadherins. We measured N-cadherin gene expression, and we obtained similar results in S_18_ and S_90_ nanopatterns, indicating that the condensates formed in these substrates presented also similar values of surface tension. Nevertheless, we observed higher stability in culture for condensates on S_90_. When the cell condensates were maintained in culture under chondrogenic induction for longer periods of time, cell condensates of S_0_ and S_18_ collapsed at day 14, while those in S_90_ retained their 3D structure (Appendix A
Appendix A).

Previous studies showed an existing interplay between cell–cell mediated cohesion, which is essentially exerted by cadherins, and the integrin-mediated cell cohesion [44]. The latter is an indirect mechanism regulated by the ECM in which with increasing integrin expression, cells lose their ability to detach from one another (through the matrix in between) or from the substrate. Since S_18_ and S_90_ condensates show a similar surface tension, given by *CDH2* expression levels, but S_90_ condensates were more stable in culture, we assume that an optimal balance between the integrin expression (integrin-mediated cell cohesion) and surface tension regulates the mechanical stability of cell condensates on the nanopatterns. Thereby, S_90_ cell condensations show higher mechanical stability than those in S_18_ due to the contribution of local surface adhesiveness, which locks cells in place and prevents them from migrating out of the condensates. As expected, this balance was completely lost in S_FN_, where higher adhesiveness hindered cell compaction (Figure 2c). This shows that local surface adhesiveness plays an active role not only during recruitment to the condensation sites, but also by enhancing condensate stability. 

The influence of adhesive crosstalk between cell–cell adhesions and cell–ECM adhesions has been previously described in early embryonic morphogenesis, where the bidirectional communication between integrins and cadherins regulates cell intercalation and migration [45]. Kalson et al. showed that as mesenchymal development progresses, cell–cell interactions become less evident in favor of ECM interactions in an environment in which the matrix is getting denser with time [46]. Cosgrove et al. found that cell–cell adhesions in MSCs caused a reduction in the mechanosensitive YAP activation, affecting the downstream signaling events. Thus, cell–cell interactions, which are progressively lost during limb development, may actuate as a regulatory mechanism to control tissue maturation [26]. 

The modification in YAP activity has been previously linked to the mechanotransduction effects on cell fate [31,47]. YAP expression is differently regulated during the chondrogenic commitment, with active YAP increasing within the first days of chondrogenesis and then decreasing during chondrocyte maturation [48,49]. Deng et al. found in micromass cultures of chondroprogenitor cells that Yap peaked at day 5 and then gradually decreased, achieving nuclear/cytosolic Yap ratios below 1 from day 10. This pattern of Yap expression coincided with that of the early chondrogenic markers Col2a1 and Col10a1 [48]. Zhong et al. found that matrix elasticity affects the fate of chondrocytes through YAP regulation. They showed that mature chondrocytes maintained their phenotype on soft substrates (4 kPa) and tended to de-differentiate on stiff substrates (40 kPa) coinciding with a higher translocation of YAP into the cell nuclei [50]. 

Similarly, we found that the levels of active YAP transcriptor coactivator were influenced by the local surface adhesiveness of the nanopatterns. YAP localization was mainly cytosolic on the nanopatterns, in agreement with previous reports of YAP activity in 3D and in which there is also a cortical actin disposition [51]. Perinuclear actin fiber contractility in 2D cultures on stiff substrates has been reported to cause nuclear flattening and the opening of nuclear pores, thereby facilitating YAP import [52]. Nevertheless, cell–cell contact activation of the Hippo pathway has been described in high-density cultures [53,54]. Accordingly, no enhanced YAP activity was found in S_FN_ controls. 

Although mainly phosphorylated and allocated in the cytosol, active YAP was found significantly higher in S_90_. This was also the only substrate in which the expression of the early chondrogenic marker *SOX9* was upregulated, thereby indicating that the local surface adhesiveness in S_90_ behaves as a positive switch for chondrogenesis and suggested that this contribution of the mechanotransduction in the chondrogenic fate is related to YAP. 

The local surface adhesiveness in S_90_ nanopatterns, in balance with cell–cell interactions, allowed the contribution of mechanotransduction in cohesiveness and YAP regulation, leading to more stable condensates able to enter the early stages of chondrogenic differentiation. On the contrary, in agreement with Cosgrove et al. [26], the low local surface adhesiveness in S_18_ nanopatterns do not compensate the cell–cell interactions, which prevent cell condensates from maturation and differentiation. In conclusion, controlling local surface adhesiveness during the first chondrogenic events is crucial to obtain stable prechondrogenic condensates and direct the fate of MSCs towards cartilage. It could be interesting to test configurations of RGD dendrimer-based nanopatterns with a higher local surface adhesiveness than S_90_ to see whether they could further improve mesenchymal condensation and chondrogenic differentiation. Since nanopatterns are obtained by endpoint equilibrium conditions, higher nanopattern densities could only be achieved by increasing the initial dendrimer concentration in solution. Previous attempts to increase dendrimer concentrations led to suspension instabilities and dendrimer aggregation [15], thus exposing some limitations to the current method. A range of dendrimer concentrations is currently being tested in our lab both in theoretical simulations and experimentally to see if it is possible to increase the local surface density, avoiding aggregate formation. In parallel, and using the S_90_ configuration as a reference, experiments with RGD-based nanopatterns on cell constructs are being developed to be tested in animal models. 

## 4. Materials and Methods 

### 4.1. Production of Nanopatterned Substrates

Previously synthesized RGD–Cys–D1 dendrimers were used [15] (Appendix A
Appendix A). All dendrimer solutions were sonicated and filtered through a Millex RB sterile syringe filter from Merk Millipore (Darmstadt, Germany prior to use, and stock solutions were used within 6 months of preparation.

Nanopatterned substrates were prepared as previously described [15,16,17]. Briefly, a 95/5 L-lactide/DL-lactide copolymer from Corbion (Montmeló, Spain) at 2% *m/v* solution in dry 1,4-dioxane from Sigma-Aldrich (Darmstadt, Germany) was spin-coated at 3000 rpm for 30 s onto 1.25 cm × 1.25 cm Corning^®^ glass microslides from Sigma-Aldrich (Darmstadt, Germany). Deionized Milli-Q water of 18 MΩ·cm from Millipore (Darmstadt, Germany) was used for rinsing samples and to prepare RGD-functionalized dendrimers’ working solutions. Aqueous solutions of RGD–Cys–D1 at 2.5 × 10^−8^ and 4 × 10^−9^% *w/w* concentrations were prepared to obtain the nanopatterns of high (S_90_) and low (S_18_) local surface adhesiveness, respectively. Spin-coated PLLA substrates were treated for 13 min under UV light and immersed in dendrimer solutions for 16 h (pH = 5.6, T = 293 K) to obtain the nanopatterns. Then, in sterile conditions, the nanopatterned substrates were rinsed with copious amounts of water and dried. Positive controls (S_FN_) were obtained by incubating spin-coated PLLA substrates with fibronectin (100 μg/mL) from bovine plasma-solution from Sigma-Aldrich (Darmstadt, Germany).

### 4.2. Cell Culture

All steps were performed in a sterile tissue culture hood, and only sterile materials, solutions and techniques were used. 

Human adipose-derived mesenchymal stem cells (hAMSCs) from ATCC, Barcelona, Spain) from one donor (38-year-old Caucasian woman) were cultured at 37 °C and 5% CO_2_. Cell growth medium consisted of MSC basal medium from ATCC (Barcelona, Spain) supplemented with MSC growth kit low serum from ATCC (Barcelona, Spain) and penicillin–streptomycin from Invitrogen (Darmstadt, Germany) 0.1% *v/v*. Cells from passages 3–4 were used. Medium was changed every 3 days. Passaging was carried out when cells reached 70–80% confluence. To induce chondrogenesis, the cells were resuspended in Chondrocyte Differentiation Tool from ATCC (Barcelona, Spain), supplemented with 0.1% *v/v* penicillin–streptomycin, and seeded at a cell density of 3000 cells/cm^2^ on the substrates.

### 4.3. Live Imaging 

hAMSCs at passage 3 were trypsinized and centrifuged, and the pellet was resuspended in MSC Basal medium without serum. Basal medium containing Hoechst 33342 from Invitrogen (Darmstadt, Germany) was added on the cell suspension (final Hoechst concentration 2 µg/mL) and mixed well by pipetting. The cell suspension was incubated at 37 °C for 3–4 min; the same volume of fetal bovine serum from Gibco (Madrid, Spain) was added on the suspension, left to stand for 1 min at room temperature and centrifuged at 300 rcf for 10 min. The resulting pellet was rinsed gently with PBS from Gibco (Madrid, Spain), resuspended in growth medium and counted, then centrifuged again and resuspended in the corresponding volume of chondrogenesis-inducing medium with penicillin–streptomycin. Cells were seeded on the substrate at a density of 30,000 cells/cm^2^ in a glass-bottom microscopy dish from VWR (Llinars del Vallès, Spain). The sample was immediately transferred to the microscope setting and pre-conditioned to 37 °C and 5% CO_2_. Imaging started an average of 10 min after seeding.

Live imaging was performed with a Nikon Eclipse Ti2-E inverted microscope from Nikon Metrology (Leuven, Belgium) with a Prime 95B sCMOS camera from Photometrics (Birmingham, UK in an Okolab Cage Incubator. A 10× image was taken every 6 min for the bright-field and blue channels, for 40 h.

### 4.4. Cell Tracking

Individual cell nuclei in live imaging stacks were tracked with the TrackMate plugin in ImageJ 1.52p software from NIH (Bethesda, MD, USA). The first 17 frames (96 min) were removed before analysis to omit cell drifting. Nuclei were detected with a LoG detector with an estimated blob diameter set to 15 µm and a threshold set to 4 µm. A LAP tracker was used for nuclei tracking, allowing frame-to-frame linking and track segment gap closing to a maximum of 50 µm and 1 frame.

Representative migratory trajectories of individual cells obtained from TrackMate were selected and normalized to start at the origin of the graph. The track displacement of all the individual cell trajectories was computed to obtain the Euclidean distance plot. Representative mean link displacements for individual cell trajectories were selected.

### 4.5. Immunostaining and Image Acquisition

For YAP and actin staining, hAMSCs at passage 3 were seeded in chondrogenesis-inducing medium on S_0_, S_18_, S_90_ and S_FN_ substrates and incubated for 6 or 72 h. For type II collagen quantification, hAMSCs at passage 4–5 were seeded in chondrogenesis-inducing medium on the substrates and incubated for 5 days (medium was replaced every second day). Three replicates were seeded for each condition in both cases. Cultured cell samples were carefully rinsed with PBS, fixed with formalin solution from Sigma-Aldrich (Darmstadt, Germany) for 20 min at room temperature and rinsed again twice with PBS. Aldehyde groups were blocked with ammonium chloride from Sigma-Aldrich (Darmstadt, Germany) 50 mM in PBS for 20 min. Samples were permeabilized with saponin 0.1% *m/v* in BSA both from Sigma-Aldrich (Darmstadt, Germany) 1% m/v in PBS for 10 min.

Samples were stained with primary antibody against YAP from Santa Cruz Biotechnology (Madrid, Spain) or against anti-collagen alpha-1 XX chain from Acris Antibodies (Madrid, Spain, in BSA 1% *m/v* PBS for 2 h at room temperature, then with secondary fluorophore-conjugated antibody anti-mouse Alexa 488 from LifeTech (Madrid, Spain) in BSA 1% *m/v* in PBS for 2 h. SiR-Actin from TebuBio (Barcelona, Spain) and Hoechst 33342 were added to the secondary antibody solution for actin and cell nuclei staining. Samples were mounted with coverslips in Fluoromount mounting medium from Sigma-Aldrich (Darmstadt, Germany). 

Samples were imaged with a Leica SPE upright confocal microscope from Leica Microsystems (Hospitalet del Llobregat, Spain with a 63X objective. The distance between imaged slices (z-size) was set at 1 μm. At least 3 representative images were taken of each sample.

### 4.6. Analysis of Actin Volume and YAP Localization

ImageJ 1.52p software software was used for image analysis and quantification. For the analysis of the volume of cell condensates, confocal images of the actin staining were used. A threshold was applied. The automatic Otsu method was selected. A pre-defined macro to calculate volume from stacks was run [55]. 

For YAP localization analysis, cell nuclei were randomly selected in each sample; YAP mean intensity was measured in an area of equal size inside and just outside each nucleus. The factor between the nuclear and the cytosolic intensity was calculated as a quantification of YAP nuclear translocation. At least 4 nuclei were quantified in each image.

### 4.7. RNA Extraction and Retrotranscription

hAMSCs at passage 3–4 were seeded in chondrogenesis-inducing medium on S_0_, S_18_, S_90_ and S_FN_ substrates. Reverse transcription real-time PCR (RT-qPCR) was performed to quantify *CDH2* and *SOX9* expression after 6 days of chondrogenesis. mRNA was extracted from the samples and purified with an RNeasy Micro Kit from Qiagen (Hospitalet del Llobregat, Spain). Extracted mRNA was quantified in a Nanodrop ND-1000 spectrophotometer from Thermo Fisher Scientific (Waltham, MA, USA). Reverse transcription for cDNA production was done with an iScript Advanced cDNA synthesis kit from Bio-Rad (El Prat del Llobregat, Spain) in a T100 thermal cycler (Bio-Rad). Three cell culture replicates of each condition were conducted, with their RNA extracted and retrotranscribed. The same amount of RNA was retrotranscribed for all samples. 

### 4.8. qPCR and Data Analysis

qPCR was performed with the Sso Advanced Universal SYBR Green Supermix kit from Bio-Rad (El Prat del Llobregat, Spain) in an Applied Biosystems StepOnePlus Real-Time PCR Machine from Thermo Fisher Scientific (Waltham, MA, USA). Commercial primer pairs were used for *CDH2* and *SOX9*, as well as *B2M* and *RPL24* all from Bio-Rad (El Prat del Llobregat, Spain) as housekeeping genes. To prevent gDNA amplification, a DNase digestion step was included during RNA extraction, and intron-spanning primer pairs were selected. The amplification program consisted of an initial activation step of 30 s at 95 °C, followed by 50 cycles of 10 s at 95 °C for denaturation and 1 min at 60 °C for annealing and extension and a final denaturation step of 15 s at 95 °C. Melt curves were performed from 65 °C to 95 °C in steps of 0.5 °C. Technical duplicates of each sample were performed in the qPCR.

qPCR data was analyzed with qBase+ software version 3.1 from Biogazelle (Zwijnaarde, Belgium). Expression of each gene was calculated by the 2−ΔΔCt method, normalized to that of S_0_ substrates (assigned value 1) and presented as relative mRNA expression levels.

### 4.9. Statistical Analysis

Quantitative data were displayed showing average and SE of the mean. N refers to number of experimental repeats of the same sample. Data were found not significantly drawn from a normally distribute population at the 0.05 level in a Shapiro–Wilk test of normality. Significant differences were judged using the one-way ANOVA and Tukey multiple comparison test, with a significance level of 0.05. 

## Figures and Tables

**Figure 1 ijms-21-05269-f001:**
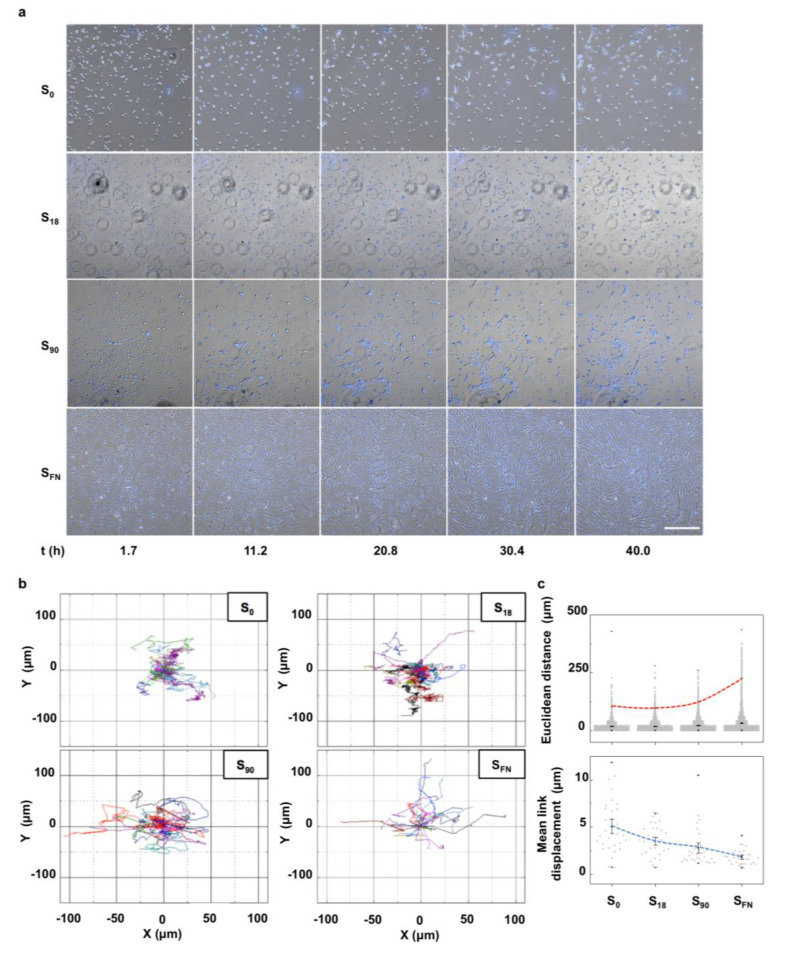
Single-cell tracking of hMSCs on nanopatterns of low (S_18_) and high (S_90_) local surface adhesiveness under chondrogenic conditions for 40 h. (**a**) Selected time-lapse bright-field images superimposed with fluorescence live images of cell nuclei (Hoechst 33342 nuclear stain). Scale bar = 500 μm. (**b**) Representative migratory trajectories for individual cells (*n* = 30). The beginning of each trajectory has been tied to the origin of the Cartesian coordinate system. Trajectories of different cells are shown in different colors. (**c**) Top: Quantification of the Euclidean distance covered by individual cells. Red dashed line indicates the evolution in the 99th percentile with the increasing local surface adhesiveness (*n* = 3000). Bottom: Quantification of mean link displacement of individual cell trajectories (*n* = 30). The blue dashed line indicates the evolution of the mean with the increasing local surface adhesiveness. Results are given as the mean ± SE.

**Figure 2 ijms-21-05269-f002:**
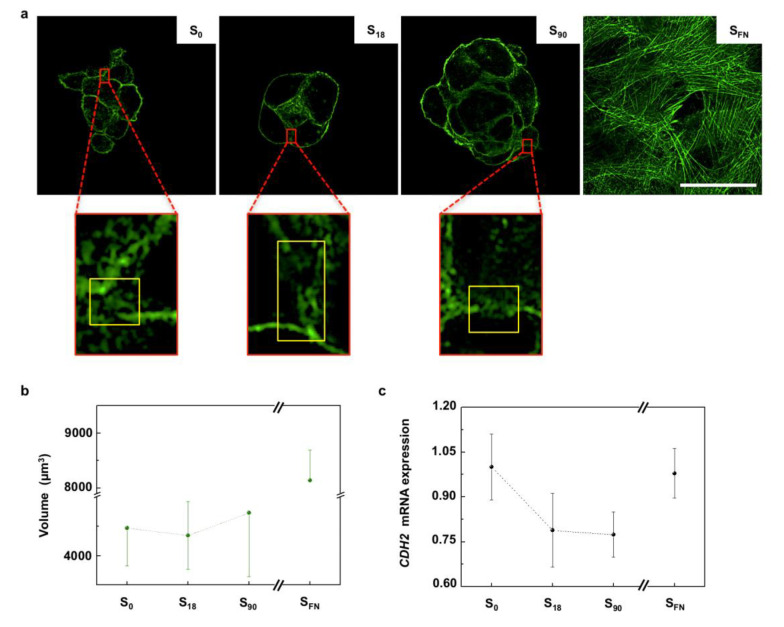
The role of cortical actin during cell condensation. (**a**) Representative confocal images showing actin cortical organization in the cell condensates grown on nanopatterns of low (S_18_) and high (S_90_) local surface adhesiveness after 6 h of chondrogenic induction. Scale bar = 50 μm. Magnified regions (×7) show representative actin clearing in the cell–cell contacts. (**b**) Plot of condensate volume obtained by actin staining quantification after 6 h of chondrogenic stimulation; *n* = 9. (**c**) Plot of the mRNA expression of N-cadherin (CDH2) obtained after 6 days of chondrogenic induction (relative to S_0_); *n* = 6. Results are given as the mean ± SE.

**Figure 3 ijms-21-05269-f003:**
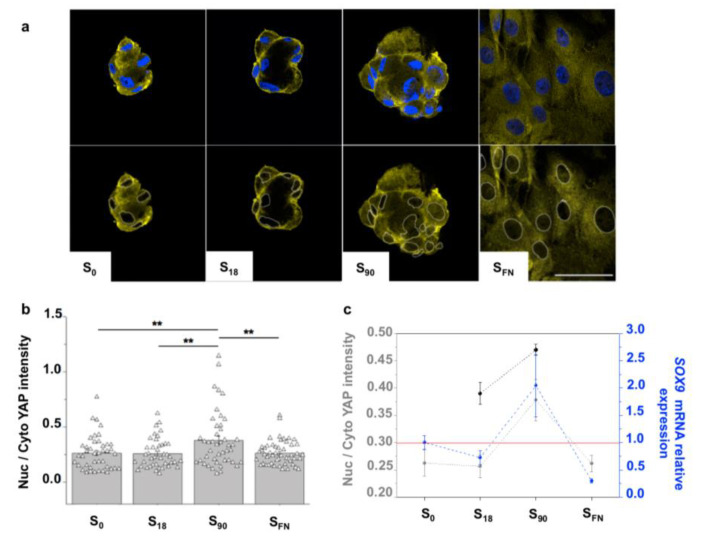
Impact of local surface adhesiveness in chondrogenesis. (**a**) Representative confocal images showing YAP localization. The upper row shows the merged nuclei (Hoechst) and YAP staining. In images in the lower row, cell nuclei perimeters are superimposed for an easy visualization of YAP allocation. Scale bar = 50 μm. (**b**) Quantification of YAP localization after 6 h of chondrogenic induction. Symbols represent each cell; *n* = 40; ** significant differences at 0.05 level. (**c**) Comparison between YAP localization at 6 h (dark grey) and at 3 days (pale grey) of chondrogenic induction with the *SOX9* mRNA expression levels (relative to S_0_) after 6 days of chondrogenic induction; *n* = 6. Results are given as the mean ± SE.

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
