# Peer review of "The Janus Role of Adhesion in Chondrogenesis"

_ijms, 2020, doi:10.3390/ijms21155269_

Round 1

Reviewer 1 Report

The manuscript entitled "The Janus role of adhesion in chondrogenesis" was interesting. Authors claimed that the adhesion of stem cells is important in chondrogenesis. The manuscript could improve by a more step-wise description of the findings and methodology used. The relevance of statements need to be more clear for enabling readers to follow the story. Especially, I am confused to read the result section. Result section is included the other research data, like discussion. Authors must describe their work and findings, not showing previous data. Overall, major revisions are required before the manuscript will be further processed.

However, major revisions are required before the manuscript will be further processed.

Line 47: ….. chondrocytes dedifferentiate during,,,,… Please add the reference. Also, most of chondrogenesis methodology is based on 3D culture. Please verify that too with proper reference.

Line 60, 82….: reference numbering format is needed to check. Also, reference style is wrong. It is not MDPI style.

Figure 1: legend of graph and picture is not readable. Also, what is the meaning of low and high local surface adhesiveness? You didn’t mention the difference between S18 and S90 on material and method section. Please clarify that and show the characteristics that divided into low and high surface adhesiveness. Also, did you coated the nanopatterned substrate?? Where? And how? Even though you used your previous technique, it is important to mention and describe shortly on material and method section.

Figure 2: Fig2a showed the cell condensates. I see the cell size difference between S0 and S18/90. It seems the stem cells on S18 and S90 were senescent compared to S0. To verify that the nanopattern particles are not harmful to stem cell, I recommend live/dead cell viability assay or trap staining. Additionally, Fig2c showed the CDH2 mRNA expression. I assumed that you collected all the cells on the dishes. How did you quantify the cell numbers??? I think that the cell numbers on each sample are quite different, as you mentioned on supplemental data. Please clarify the quantification.

Line 297: information of cell culture is needed. Cell numbers and passage, and did you use serum or not for chondrogenic differentiation?

Figure 3: in the chondrogenesis, E-cadherin expressions is critical to local adhesive characteristics in the cell-cell interaction. I recommend E-cadherin staining.

Line 369: you mentioned the B2M and RPL24 genes on PCR data. However, I don’t see any result. If you have, you should include.

Overall, the article is well performed to show the adhesive effect on chondrogenic potentials of stem cells. However, the figure presentation and result description must be improved. Additionally, there is no chondrogenic marker expression added. If authors want to convince the adhesive effect of nanopattern substrates on chondrogenesis, at least col2 expression must be presented or alcian blue staining.  

Reviewer 2 Report

This article presents an interesting investigation of adhesive RGD peptide density on the cartilage formation potential of MSC condensates, and provides insight into the relationship between YAP localization and cartilage differentiation progression. There are some concerns about the novelty of the approach, as quite a few papers in the past have looked at factors such as RGD density and alternative approaches to induce condensation of MSCs during chondrogenesis that would be likely to result in a similar pattern of cell-cell binding vs. cell-matrix binding. There are also a number of typos and spelling errors throughout the manuscript, a few of which are mentioned below. Despite these concerns, overall the manuscript is fairly well written and would likely be of interest to the readership of IJMS, and therefore is suggested for acceptance with revision.

Comments:

-It is not clear why the nano patterning approach is believed to be preferential to other methods of altering the RGD peptide density on the surface.

-There are alternative methods of inducing cell condensation (e.g. centrifugation, micro-well plates) that cause similar types of cell-cell adhesion changes and chondrogenic differentiation. Although using something like this as a control would be ideal, it is recommended to at least provide some additional discussion on this topic.

-Although they reference prior work, data on the dendrimer characterization would be helpful to include in the manuscript. 

-In this reviewer’s opinion, the surface characterization data would be helpful to move into a primary figure rather than supplemental.

-The text in the figures is pixelated and hard to read in the manuscript pdf.

-Figure 3. Please also include panels showing the actual nuclei staining in addition to the outlines in the YAP data (Fig. 3a).

-Since s90 shows the strongest response in inducing cartilage differentiation of the cells, why not go higher in density? Where would the maximum response be expected? 

-Figure 3 caption. “Cell nuclei perimiters” —> “Cell nuclei perimeters”

-Line 242. “atributed” —> attributed 

-Line 242. “higer” —> higher

-There are likely other typos that are not listed here, please take a close read.

-It is not clear what advantage the dendrimer preparation provides over a standard surface coating technique with RGD peptide, which has been used in the past in quite a few papers. 

Reviewer 3 Report

Casanellas et al. investigated the connections between pre-condensation of ASC chondrogenesis and the local surface adhesiveness tuned by RGD dendrimer nanopatterns. Specifically, the authors put efforts on cell migration, actin reorganization and YAP localization during the chondrogenesis of ASCs. This manuscript is well written, and experiment is well designed. Results are robust and well presented. The discussions are somewhat relevant with quite a few references. However, some questions need to be addressed to distinguish this manuscript from the authors’ previous work. In addition, several concerns are listed below to improve the quality of this study:

  1. Why did the authors choose the S90 and S18 RGD content instead of using other ratios? Please clarify and explain.
  2. Honestly, I would not rely on just SOX9 mRNA expression for the evaluation of chondrogenesis. Can the author perform more staining or markers for chondrogenesis? Such as collagen, Safranin O staining, GAG content/cell. (I notice the authors have tested COL2A1 in other publication and I know this study is a different strategy as no centrifuge involved).
  3. I feel the data is actually pretty good (format needs some revisions though). However, I was a little confused what exactly the authors are trying to elucidate in this manuscript compared to their previous studies (DOI 10.1007/s12274-016-1382-5; DOI: 10.3791/56347). To me, the methods are the same, the groups are the same, the time points are the same and the markers are the same except the YAP localization. And the interesting part is the authors really did not discuss these two references. I would like to see some comments from the authors for the elevation of the whole study and how it connects previous work.
  4. A follow up comment: the results section involved too much discussion and references and really dilute the significance of the “real” discussion part. I think the authors can rearrange some of the statements and put them into discussion section.

Minor:

  1. I feel some of the discussion are not that relevant (e.g the stiffness of substrate).
  2. Since the authors called the FN group a ‘positive control’, I would assume that this is only the positive control for adhesiveness instead of cellular fate. Because from the SOX9, clearly FN is not a good positive group of chondrogenesis. Or the FN can only be a positive control for the pre-chondrogenesis.
  3. Labels for figure 1 is too small to read (e.g S0, S18, S90 and SFN; the y axis for Figure 1c, and this applies to other figures).
  4. Figure S1c, RGD content number missing “x”.
  5. It is always helpful to double check the language and grammar errors.

Reviewer 4 Report

In the submitted manuscript, the authors analyzed the performance of RGD nanopatterns of low (S18) and high (S90) local surface adhesion for cartilage tissue engineering. They evaluated the effect of these nanopatterns on prechondrogenic condensation of human adipose-derived stromal cells (ASC) on the substrate by fluorescence live cell imaging, immunostaining, confocal microscopy, and semi-quantitative RT-PCR. They found that S90 nanopatterns (90% of the surface area with nanospacing below 70nm) induced increased adhesion, more directional migration and better consensate stability than S18 nanopatterns, thus favoring chondrogenic commitment.

This is a work with of a very interesting topic, that is the very challenging issue of mesenchymal stromal cell differentiation towards a hyaline cartilage phenotype through substrate modulation at the nanoscale level.

The paper is well written, and experiments appear well performed, however there are some criticisms needing clarification.

  • In the abstract (lane 28) and in the discussion (lanes 279-280) authors speak about viable condensates, but no data are provided about cell viability. Data point to increased condensate formation, organization, and stability on S90 substrate, but not viability. Do the number and the volume of condensates depend on cell viability? This should be better explained, or, in alternative, viability data should be provided.

  • How many samples were analyzed? It seems to me that only one type of sample (ASC from a single cell preparation) was used with a certain number of replicates. If, on the contrary, different ASC preparations (I mean from different donors) were used, please indicate how many different samples were analyzed. Please, specify if N indicates the number of different samples analyzed or the number of experimental replicates.

  • In relationship to the previous comment, there is an issue with statistical analysis. Authors applied one way-Anova, but this can only be done if different groups are compared. Technical replicates of the same sample cannot be considered as a group. In case of repeated analysis of only one sample (ASC from one donor), only descriptive statistics should be provided.

  • Only SOX9 mRNA expression was analyzed to support initial chondrocyte differentiation, based on its precocious expression during this process. This limits the relevance of the result and additional chondrocyte markers could be evaluated, maybe at the more advanced times of observation (day 14). Moreover, PCR conditions to detect SOX9 mRNA include 50 amplification cycles (lane 372), that is a very high number, raising the risk to detect biologically irrelevant transcript levels. Why so many cycles were used? Could you provide mean Ct obtained for housekeeping genes and for SOX9? This would help in demonstrating consistence of the data.

  • In general, results are structured as discussion sections, with several comments and citations (see for example lines 138-144; 184-201; 210-227). This raises some confusion between presented data and data obtained in previous work, therefore I suggest reorganizing the text to provide all the already published data in the Introduction or in the Discussion. Only data obtained in the present work should be presented in the Result section.

Minor points:

  • Figure 3c is not clear. Unfortunately, I cannot appreciate the difference between pale and dark grey, therefore colors should be changed, and the different parameters showed in the figure should be better identified.

Round 2

Reviewer 1 Report

The manuscript has been well-revised and improved. However, the cell viability and chondrogenic differentiation are not demonstrated in revised manuscript. I stronly recommend the cell/dead viability staining. I am not sure about the cell viability becaused of cell size between the groups. Senescent cells show Also, include the type II collagen staining image in Figure S4, please.

Author Response

ijms-829530

"The Janus role of adhesion in chondrogenesis"

Answers to Reviewers

The authors are very thankful to the Reviewers for their comments on the manuscript. Listed below are the answers to the Reviewers questions. The corresponding changes have been included in the revised version of the manuscript and highlighted in the text.

Reviewer #1

The manuscript has been well-revised and improved. However, the cell viability and chondrogenic differentiation are not demonstrated in revised manuscript. I strongly recommend the cell/dead viability staining. I am not sure about the cell viability because of cell size between the groups. Senescent cells showed. Also, include the type II collagen staining image in Figure S4, please.

As requested, cell/dead viability staining was performed and included in the revised version of the manuscript (Figure S3, highlighted). Regarding, type II collagen staining, it was included as Figure S4 in the revised version of the manuscript (IJMS_Supporting Information_Revised.docx). Some issue had probably occurred during the uploading process. A notification to the Editor has been send regarding this question. In any case, a new file containing also Figure S3 for the cell viability will be uploaded in the new revised version of the manuscript.

Reviewer 2 Report

The authors responded to many of the concerns of this reviewer. If they address the following comments, in the eyes of this reviewer the paper could be potentially suitable for publication.

Comment 1: “It is not clear why the nano patterning approach is believed to be preferential to other methods of altering the RGD peptide density on the surface.” The authors provide a convincing response in the cover letter, however most of the material was not incorporated into the text of the manuscript. Please add to the discussion.

Comment 7. “Since S90 shows the strongest response in inducing cartilage differentiation of the cells, why not go higher in density? Where would the maximum response be expected?” 

The author response to this concern is ok, but please include a statement in the discussion about the current experimental limitations of dendrimer concentration and what the expected cell findings could be with higher surface density.

Author Response

ijms-829530

"The Janus role of adhesion in chondrogenesis"

Answers to Reviewers

The authors are very thankful to the Reviewers for their comments on the manuscript. Listed below are the answers to the Reviewers questions. The corresponding changes have been included in the revised version of the manuscript and highlighted in the text.

Reviewer #2

The authors responded to many of the concerns of this reviewer. If they address the following comments, in the eyes of this reviewer the paper could be potentially suitable for publication.

Comment 1: “It is not clear why the nano patterning approach is believed to be preferential to other methods of altering the RGD peptide density on the surface.” The authors provide a convincing response in the cover letter, however most of the material was not incorporated into the text of the manuscript. Please add to the discussion.

As requested by the Reviewer, the advantages of the nanopatterning approach with respect to other methods of altering the RGD peptide density have been included in the discussion section in the new revised version of the manuscript (highlighted).

Comment 7. “Since S90 shows the strongest response in inducing cartilage differentiation of the cells, why not go higher in density? Where would the maximum response be expected?” The author response to this concern is ok, but please include a statement in the discussion about the current experimental limitations of dendrimer concentration and what the expected cell findings could be with higher surface density.

As requested by the Reviewer, a statement in the discussion about the possibility of increasing dendrimer concentration and the current experimental limitations encountered has been added to the discussion section in the new version of the revised manuscript (highlighted).

Reviewer 3 Report

Great work!

Author Response

Thanks

Reviewer 4 Report

A more detailed indication of corrections (line numbers and maintenance of deleted lines) would have helped.

1.Viability test was not provided although the authors agree viability can influence condensate number and volume, therefore it appears a relevant parameter to be provided here and not in the future.

2 and 3. “Most of the cells are from the same commercial vial” is an unspecific answer. How many cell preparations (I mean from different donors) were used? Which cell preparations in which experiments? The authors must indicate the precise number of different cell preparations (different donor origin) used in each experiment.

Data obtained from the same cells (same donor origin) are expected to be more similar than data obtained from different cell preparations (different donor origin) and cannot be included in the same group of statistical analysis without raising a bias (in this case each group would be composed of some “repeated” samples and some unique samples and this is not correct). The same cell preparation tested several times represents a repeated experiment inside each group (control group, S18 group, S90 group…) and this affect statistical analysis since samples from the same cell preparation are not independent samples.

Conversely, if cell preparations from a single donor sample were used, they are likely to be homogeneous for biological properties and I agree that this increases the chance to detect differences dependent on other variables.

Other issues about statistical analysis: - one-way Anova is a parametric test that necessitates to verify normal distribution of the data. This should be done and described; -multiple comparisons as those performed in the study need post-hoc analysis and correction of the p cut-off. This correction must be performed.

  1. Technical details about collagen II expression analysis by confocal microscopy are needed (antibody used, conditions, etc). No information was added in the materials and methods, but results are described.
  2. In my opinion the result section still contains too much literature data and should be further reduced in order to be limited to results of the present work.

Author Response

ijms-829530

"The Janus role of adhesion in chondrogenesis"

Answers to Reviewers

The authors are very thankful to the Reviewers for their comments on the manuscript. Listed below are the answers to the Reviewers questions. The corresponding changes have been included in the revised version of the manuscript and highlighted in the text.

Reviewer #4

A more detailed indication of corrections (line numbers and maintenance of deleted lines) would have helped.

We apologize for that. Authors thought that the maintenance of the deleted text when restructuring large parts of the manuscript could be confusing, therefore we preferred to just highlight the changes made instead.

1.Viability test was not provided although the authors agree viability can influence condensate number and volume, therefore it appears a relevant parameter to be provided here and not in the future.

According to the Reviewer indications, cell/dead viability staining was performed and included in the revised version of the manuscript (Figure S3, highlighted).

2 and 3. “Most of the cells are from the same commercial vial” is an unspecific answer. How many cell preparations (I mean from different donors) were used? Which cell preparations in which experiments? The authors must indicate the precise number of different cell preparations (different donor origin) used in each experiment.

We used only one donor for the reported experiments:

PSC-500-011 Adipose-Derived Mesenchymal Stem Cells; Normal, Human

Lot # 60866843 from ATCC Primary Cell Solutions.

According to the certificate of analysis (attached), the vial was obtained from a liposuction of a Caucasian female of 38 years old.

This was specified in the Materials and Methods section and highlighted in the new revised version of the manuscript.

Data obtained from the same cells (same donor origin) are expected to be more similar than data obtained from different cell preparations (different donor origin) and cannot be included in the same group of statistical analysis without raising a bias (in this case each group would be composed of some “repeated” samples and some unique samples and this is not correct). The same cell preparation tested several times represents a repeated experiment inside each group (control group, S18 group, S90 group…) and this affect statistical analysis since samples from the same cell preparation are not independent samples. Conversely, if cell preparations from a single donor sample were used, they are likely to be homogeneous for biological properties and I agree that this increases the chance to detect differences dependent on other variables.

We agree with the Reviewer in that since the cells used are from the same donor, the groups compared are not biological replicates. However, as pointed by the Reviewer, cell preparations from the same donor are more likely to be homogeneous for biological properties, and as mentioned, since we are testing the difference introduced by the local surface adhesiveness of the substrates in cell behavior, we tried to minimize all the other sources of variability that could influence the results. Normally, in these cases the influence of the donor was not considered unless a feature related to donor variability was expected to influence the experiment. In a recent publication that we performed in collaboration with clinicians, we evaluated the effects of RGD-based nanopatterns on human mesenchymal cells derived from osteoarthritic and healthy donors.[1]

Other issues about statistical analysis: - one-way Anova is a parametric test that necessitates to verify normal distribution of the data. This should be done and described; -multiple comparisons as those performed in the study need post-hoc analysis and correction of the p cut-off. This correction must be performed.

All the data subjected to one-way ANOVA test was not significantly drawn from a normal distribution at the 0.05 level in a Shapiro-Wilk test. Also, Tukey test was performed after the ANOVA for multiple comparisons. This and the p cut-off have been indicated in the new revised version of the manuscript (highlighted).

Technical details about collagen II expression analysis by confocal microscopy are needed (antibody used, conditions, etc). No information was added in the materials and methods, but results are described.

Technical details regarding type II collagen quantification have been added to the Materials and Methods section of the new revised version of the manuscript (highlighted).

In my opinion the result section still contains too much literature data and should be further reduced in order to be limited to results of the present work.

We understand the Reviewer concern, but the Authors feel that by completely removing the background from the Results section, the line of argument could be compromised. Therefore, we prefer not to modify it.

[1] Rodríguez-Pereira, C.; Lagunas, A.; Casanellas, I.; Vida,Y.; Pérez-Inestrosa, E.; Andrades, J.A.; Becerra, J.; Samitier, J.; Blanco, F.J.; Magalhães, J. RGD-Dendrimer-Poly(L-lactic) Acid Nanopatterned Substrates for the Early Chondrogenesis of Human Mesenchymal Stromal Cells Derived from Osteoarthritic and Healthy Donors. Materials, 2020, 13, 2247.

Round 3

Reviewer 1 Report

The manuscript was well-described and explained about author's experiment. The authors greatly improved the manuscript by implementing the recommendations of the reviewers.

Author Response

Thank you

Reviewer 4 Report

In the present version of the manuscript, authors provided the requested viability test, collagen II analysis information, and they declared the use of a single donor cell preparation.

I am still skeptical about statistical analysis. A post hoc test is declared, and this generally requires a p cut-off reduction according to the number of comparisons, but the p cut-off was maintained at 0.05 (as stated at line 420). Since I am not specifically competent, I strongly suggest a check by a statistician. Moreover, N is not the sample size (lines 417-8), but the number of experimental repeats of the same sample: please, change the text accordingly.

I have no additional comments.

Author Response

In the present version of the manuscript, authors provided the requested viability test, collagen II analysis information, and they declared the use of a single donor cell preparation.

I am still skeptical about statistical analysis. A post hoc test is declared, and this generally requires a p cut-off reduction according to the number of comparisons, but the p cut-off was maintained at 0.05 (as stated at line 420). Since I am not specifically competent, I strongly suggest a check by a statistician. Moreover, N is not the sample size (lines 417-8), but the number of experimental repeats of the same sample: please, change the text accordingly.

We thank the Reviewer to rise this important point, we were misusing the p-value term, which as noted by the Reviewer depends on the number of comparisons. We were referring to the level of significance (α), which was set to 0.05. Correction has been made to the new revised version of the manuscript and highlighted in the text. Also, N was indicated as the number of experimental repeats of the same sample as requested.

I have no additional comments.
